# From Out-of-Distribution Detection to Hallucination Detection: A Geometric View

Litian Liu[1]  Reza Pourreza[1]  Yubing Jian[1]  Yao Qin[2]  Roland Memisevic[1]

## Abstract

Detecting hallucinations in large language models is a critical open problem with significant implications for safety and reliability. While existing hallucination detection methods achieve strong performance in question-answering tasks, they remain less effective on tasks requiring reasoning. In this work, we revisit hallucination detection through the lens of out-of-distribution (OOD) detection, a well-studied problem in areas like computer vision. Treating next-token prediction in language models as a classification task allows us to apply OOD techniques, if we bring to bear appropriate modifications to account for the structural differences in large language models. We show that approaches based on OOD detection yield training-free, single-sample based detectors, achieving strong accuracy in hallucination detection in reasoning tasks. Overall, our work suggests that reframing hallucination detection as OOD detection provides a promising and scalable pathway toward language model safety.

## 1. Introduction

Hallucination remains a fundamental obstacle to the reliable deployment of Large Language Models (LLMs), and their detection has therefore been an important research area in recent years. One prominent line of work (Azaria & Mitchell, 2023; Kossen et al., 2024; Li et al., 2023a; Liu et al.; Manakul et al., 2023; Marks & Tegmark; Su et al., 2024; Wang et al., 2024) trains a separate classifier to detect hallucinations. These methods can be sensitive to distribution shifts and incur high training costs. Training-free methods (Azaria & Mitchell, 2023; Kossen et al., 2024; Li et al., 2023a; Liu et al.; Manakul et al., 2023; Marks & Tegmark; Su et al.,

2024; Wang et al., 2024), by contrast, measure discrepancies across multiple samples to detect hallucinations. While they avoid training overhead, they impose significant computational costs during inference—particularly in multi-step reasoning. Furthermore, although such methods perform well on concise question-answering tasks, they face challenges in reasoning, where the inherent diversity of valid reasoning paths makes comparing multiple outputs conceptually difficult. This motivates the need for *training-free*, *single-sample* based hallucination detection algorithms within the increasingly critical domain of reasoning.

Conceptually, hallucination detection is reminiscent of out-of-distribution (OOD) detection (Hendrycks et al., 2019; Liang et al., 2018; Liu et al., 2020; Sun et al., 2021; Sun & Li, 2022; Xu et al., 2024; Lee et al., 2018; Sun et al., 2022; Liu & Qin, 2024; 2025), a dominant research area for classification tasks. In OOD detection, the goal is to identify test samples whose class labels were not seen during training. For such samples, the model will still produce a prediction from the set of training classes, despite having no knowledge of the true label—effectively generating a "hallucination" of the classifier. At their core, both OOD detection and hallucination detection in LLMs boil down to measuring the model's uncertainty (Kuhn et al., 2023b; Liu & Qin, 2024).

A natural question therefore is: can we leverage the rich literature on OOD detection to address the challenge of hallucination detection? Unfortunately, despite the conceptual similarities, several practical challenges arise when applying OOD detection methods to hallucination detection in LLMs. First, many OOD detectors rely on the empirical estimation of training statistics (Wang et al., 2022; Liu & Qin, 2025; Hendrycks et al., 2022; Lee et al., 2018; Liu et al., 2023) such as the mean of penultimate-layer features. While these statistics effectively characterize the training distribution and enable OOD detection, directly estimating them from LLM training data is impractical. Modern training corpora are often proprietary, and even when accessible, their scale renders computation prohibitively expensive. Second, the label space expands dramatically when moving from conventional classifiers to LLMs. OOD scores designed for small label spaces can become noisy and less reliable, and the expanded space furthermore amplifies computational costs,

[1]Qualcomm AI Research. Qualcomm AI Research is an initiative of Qualcomm Technologies, Inc. [2]UC Santa Barbara. Correspondence to: Litian Liu <litiliu@qti.qualcomm.com>.

*Proceedings of the 43rd International Conference on Machine Learning*, Seoul, South Korea. PMLR 306, 2026. Copyright 2026 by the author(s).

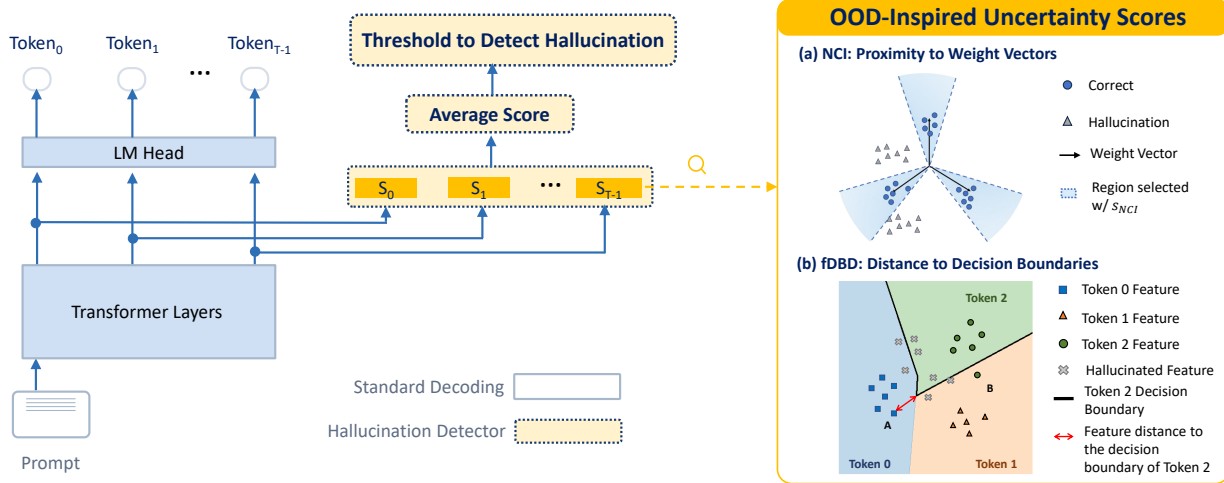

*Figure 1.* **Overview of OOD-Inspired Hallucination Detectors.** At each decoding step, we compute an OOD-inspired uncertainty score, average it across steps, and apply a threshold to detect hallucinations; higher uncertainty indicates a higher likelihood of hallucination. We demonstrate this framework with two geometric scores adapted to LLMs: (a) feature proximity to class weight vectors (Definition 3.1) based on the OOD detector NCI (Liu & Qin, 2025), where lower proximity indicates higher uncertainty. (b) distance to the decision boundary (Definition 3.2) based on the OOD detector fDBD (Liu & Qin, 2024), where smaller distance indicates higher uncertainty.

particularly in multi-step reasoning tasks where expenses accumulate across decoding steps. Third, unlike classifiers, which typically make deterministic decisions, LLMs are generative and may employ stochastic decoding. The inherent variance from token sampling therefore makes it unclear to what degree OOD scores are applicable in hallucination detection.

In this work, we take a step towards bridging the gap between OOD detection and hallucination detection. Since the cumulative computational cost in applying the score at each decoding step is high, we explore the adaptation of two light-weight OOD detectors, NCI (Liu & Qin, 2025) and fDBD (Liu & Qin, 2024). Both methods take a geometric view of uncertainty. NCI measures the proximity of penultimate-layer feature to weight vectors of the last layer (see Figure 1(a)), where lower proximity indicates higher uncertainty). fDBD measures the distance of penultimate-layer features to the decision boundaries (see Figure 1(b)), where smaller distance indicates higher uncertainty. To adapt these methods to solve the task of hallucination detection, we derive an analytical proxy for the required training statistics in their score function - the mean of training features in this case. In addition, for fDBD to be efficient and effective in the large label space, we explore the reduction of the distance computation to a selected subset of high-likely alternative tokens in hallucination detection.

We further show that these methods extend naturally to sequences by averaging step-wise uncertainty scores to obtain a sequence-level hallucination detection score. Experiments show that the resulting methods perform well under both

greedy and stochastic decoding.

We perform extensive experiments to validate the effectiveness of OOD-inspired hallucination detection across different types of reasoning tasks, model architectures and model scales, showing consistent superior performance compared to a range of baselines. This shows that framing hallucination detection as OOD detection provides a promising and scalable path toward improved language model safety.

## 2. Problem Statement

Let $f$ denote a large language model (LLM). Given an input prompt $\boldsymbol{x}$, let $\boldsymbol{y} = (y_1, \ldots, y_T)$ be the sequence of tokens generated by the model. We denote the set of all factually correct responses to $\boldsymbol{x}$ as $\mathcal{Y}^*(\boldsymbol{x})$. We say that the model $f$ is *hallucinating* if the generated response $\boldsymbol{y}$ is incorrect, i.e., $\boldsymbol{y} \notin \mathcal{Y}^*(\boldsymbol{x})$.

During inference, the ground-truth set of correct responses $\mathcal{Y}^*(\boldsymbol{x})$ for a given prompt $\boldsymbol{x}$ is unknown. The task of hallucination detection is to construct a **binary detector** $\mathcal{I}$ that predicts whether $\boldsymbol{y} \in \mathcal{Y}^*(\boldsymbol{x})$ based on the prompt $\boldsymbol{x}$, the response $\boldsymbol{y}$, and the model $f$'s internal representations:

$$\mathcal{I}(\boldsymbol{x}, \boldsymbol{y}; f) = \begin{cases} 1 \text{ (Hallucination)}, & \text{if } S(\boldsymbol{x}, \boldsymbol{y}; f) \leq \tau, \\ 0 \text{ (Not Hallucination)}, & \text{otherwise.} \end{cases}$$

$$(1)$$

where $S(\boldsymbol{x}, \boldsymbol{y}; f)$ is a scoring function and $\tau$ is a threshold. Under this formulation, a lower score $S$ typically reflects lower model certainty and therefore a higher likelihood of hallucination.

# 3. Geometry of Token Generation under a Classification View

In this section, we adapt two geometric measures—feature proximity to weight vectors and feature distance to decision boundaries—from the out-of-distribution detection literature for token generation. Both measures characterize the interplay between features and language head but were shown to exhibit different empirical strength across different OOD settings. We then discuss and empirically validate their connection to hallucination detection.

## 3.1. Classification View

Consider a large language model (LLM) $f$ with model dimension $d_{\text{model}}$ and vocabulary $\mathcal{V}$. At decoding step $t$, given an input prompt $\boldsymbol{x}$ and previously generated tokens $\boldsymbol{y}_{<t} = (\boldsymbol{y}_1, \ldots, \boldsymbol{y}_{t-1})$, the model produces a penultimate-layer feature $\boldsymbol{z}_x^t \in \mathbb{R}^{d_{\text{model}}}$. This representation is mapped by the language head $f_{\text{head}}$ which generates logits over the vocabulary $\mathcal{V}$.

Conceptually, the language head $f_{\text{head}}$ can be viewed as a high-dimensional linear classifier over the vocabulary $\mathcal{V}$, whose parameters are shared across decoding steps. Since $f_{\text{head}}$ is linear, given feature $z$, during greedy decoding the next token to be generated is

$$\hat{c} = \arg\max_{v \in \mathcal{V}} \boldsymbol{w}_v^\top \boldsymbol{z} + b_v.$$

Here, $\boldsymbol{w}_v$ and $b_v$ denote the weight vector and bias associated with token $v$. The next token may differ from this definition under stochastic (non-greedy) decoding, as discussed in 4.4.

## 3.2. Feature Proximity to Weight Vectors

For OOD detection, Liu & Qin (2025) propose measuring model uncertainty based on the proximity of features to class weight vectors, introducing a Neural Collapse-inspired detector termed **NCI**. Following their framework, we define the proximity of a feature $\boldsymbol{z}$ to the weight vectors of an LLM $f$ as the scalar projection of the weight vector $\boldsymbol{w}_{\hat{c}}$ onto the centered feature $\boldsymbol{z} - \boldsymbol{\mu}_G$, where $\hat{c}$ corresponds to the most confident token. Here, $\boldsymbol{\mu}_G$ denotes the estimated mean of training features; details on its estimation in LLM are discussed in Section 4.2.

**Definition 3.1** (Feature Proximity to Weight Vectors). To adapt OOD detection setting based on NCI (Liu & Qin, 2025) to hallucination detection, we define the score $s_{\text{NCI}}$ measuring the proximity of a feature $\boldsymbol{z}$ on LLM $f$ as:

$$\mathbf{s}_{\text{NCI}}(\boldsymbol{z}) = cos(\boldsymbol{w}_{\hat{c}}, \boldsymbol{z} - \boldsymbol{\mu}_G)\|\boldsymbol{w}_{\hat{c}}\|_2, \qquad (2)$$

where $cos(\boldsymbol{w}_c, \boldsymbol{z} - \boldsymbol{\mu}_G) = \frac{(\boldsymbol{z} - \boldsymbol{\mu}_G) \cdot \boldsymbol{w}_c}{\|\boldsymbol{z} - \boldsymbol{\mu}_G\|_2 \|\boldsymbol{w}_c\|_2}$.

A higher $s_{\text{NCI}}$ indicates closer proximity to the weight vectors (see Figure 1(a)), corresponding to lower uncertainty. We remark that computing $s_{\text{NCI}}$ per step requires $O(d_{\text{model}})$ operations.

## 3.3. Feature Distance to Decision Boundaries

As an alternative to NCI, Liu & Qin (2024) propose measuring model uncertainty based on feature distance to decision boundaries and introduce a fast decision-boundary-based OOD detector (**fDBD**).

In the following, we also explore adapting fDBD to hallucination detection, which can show stronger performance in some setting as we shall show. To this end, we define the region in the space of penultimate-layer representations, in which a token $c \in \mathcal{V}$ attains the maximum logit, as

$$\mathcal{R}_c = \left\{ \boldsymbol{z} \in \mathbb{R}^{d_{\text{model}}} \;\middle|\; \arg\max_{v \in \mathcal{V}} \boldsymbol{w}_v^\top \boldsymbol{z} + b_v = c \right\}.$$

We refer to this region as the *decision region* of token $c$, and to its corresponding boundary as the *decision boundary* of token $c$, by analogy with multi-class classification. Under greedy decoding, this terminology is exact; we retain the same nomenclature when stochastic decoding is employed.

The distance from a feature $\boldsymbol{z}$ to the decision boundary of an token $c \neq \arg\max_{v \in \mathcal{V}} \boldsymbol{w}_v^\top \boldsymbol{z} + b_v$ is:

**Definition 3.2** (Distance to Decision Boundary). Adapted from (Liu & Qin, 2024).

$$D_f(\boldsymbol{z}, c) = \inf_{\boldsymbol{z}' \in \mathcal{R}_c} \|\boldsymbol{z} - \boldsymbol{z}'\|_2.$$

This distance, illustrated in Figure 1(b), corresponds to the minimal perturbation required to change the model's most confident token from $\hat{c} = \arg\max_{v \in \mathcal{V}} \boldsymbol{w}_v^\top \boldsymbol{z} + b_v$ to $c \in \mathcal{V}, c \neq \hat{c}$. A larger distance corresponds to lower uncertainty.

We exclude token $\hat{c}$, the most confident token, from Definition 3.2, as the feature $\boldsymbol{z}$ already resides in the decision region of $\hat{c}$ (and the distance is zero), carrying no uncertainty information. For other tokens $c \neq \hat{c}$, to avoid iteratively solving the optimization to measure the distance, we derive an efficient approximation by relaxing the decision region.

**Theorem 3.3** (Approximate Distance to Decision Boundary). *Adapted from (Liu & Qin, 2024). Given feature $\boldsymbol{z}$ and token $c \in \mathcal{V}, c \neq \arg\max_{v \in \mathcal{V}} \boldsymbol{w}_v^\top \boldsymbol{z} + b_v$, $D_f(\boldsymbol{z}, c)$ is lower bounded by*

$$\tilde{D}_f(\boldsymbol{z}, c) := \frac{\left| (\boldsymbol{w}_{\hat{c}} - \boldsymbol{w}_c)^\top \boldsymbol{z} + (b_{\hat{c}} - b_c) \right|}{\|\boldsymbol{w}_{\hat{c}} - \boldsymbol{w}_c\|_2}. \qquad (3)$$

*See proof in Appendix I.*

For each token $c$, computing the distance requires $O(d_{\text{model}})$ operations: $O(1)$ for the logit difference in the numerator and $O(d_{\text{model}})$ for the norm computation in the denominator.

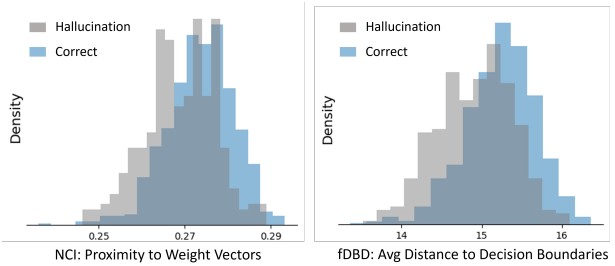

*Figure 2.* **OOD-inspired uncertainty measures signal hallucinations.** *Left:* Features from hallucinated responses exhibit lower proximity to weight vectors, extending OOD detector NCI (Liu & Qin, 2025). *Right:* Features from hallucinated responses exhibit smaller distance to decision boundaries, extending OOD detector fDBD (Liu & Qin, 2024). Experiments on the CSQA dataset with `Llama-3.2-3B-Instruct`.

### 3.4. Geometric Uncertainty Signals Hallucination

We empirically validate these two uncertainty measures in the context of hallucination detection in Figure 2. Their adaptation to LLMs is described in Section 4.2.

As shown in Figure 2 *Left*, features from hallucinated responses exhibit lower proximity to their corresponding weight vectors compared to correct responses, validating the effectiveness of **NCI** for hallucination detection. Furthermore, Figure 2 *Right* shows that penultimate-layer features from hallucinated responses lie closer to decision boundaries than those from correct answers. This observation suggests that the uncertainty signal derived from the **fDBD** detector remains a robust indicator of hallucination.

This geometric interpretation is consistent with our prior work (Liu et al., 2026), which shows that hallucinated responses are more sensitive than correct responses to intermediate-layer noise injection. Rather than relying on noise injection to implicitly probe boundary proximity through changes in model outputs, our OOD-inspired **fDBD** score explicitly quantifies distance to the decision boundary at the penultimate layer.

Overall, the empirical separation observed for both **NCI** and **fDBD** suggests that uncertainty measures from OOD detection transfer well to the hallucination detection setting. We next discuss the design of hallucination detectors that leverage and adapt these geometric properties.

## 4. From OOD to Hallucination Detection

While both OOD detection and hallucination detection are fundamentally linked to model uncertainty, the two problems differ significantly in scale and mechanism. In this section, we describe how we bridge these challenges and adapt methods originally developed for OOD detection to the context of hallucination detection.

### 4.1. Case Study Setups

In this case study, we examine hallucination detection on CSQA (Talmor et al., 2019) using `Llama-3.2-3B-Instruct` (Grattafiori et al., 2024). CSQA evaluates commonsense reasoning in a multiple-choice format, and we report results on the validation set, which contains 1,319 questions. We prompt the model using in-context learning examples following Wei et al. (2022). Details of the prompting strategy and answer extraction procedure are provided in Appendix G. Unless otherwise specified (Section 4.4), we use greedy decoding (i.e., $temp = 0$). Following Chen et al. (2024); Kuhn et al. (2023a); Lin et al. (2023; 2022b); Malinin & Gales (2020), we quantify hallucination detection performance using the threshold-free metric area under the receiver operating characteristic curve (AUROC), where higher values indicate better performance. As a baseline, we consider the model's Perplexity (Ren et al., 2023), defined as

$$\text{PPL}(\boldsymbol{y} \mid \boldsymbol{x}) = \exp\left(-\frac{1}{T}\sum_{t=1}^{T} \log p(\boldsymbol{y}_t \mid \boldsymbol{x}, \boldsymbol{y}_{<t})\right).$$

Lower perplexity corresponds to higher model confidence in the generated output. To ensure consistency with Equation 1, we negate the perplexity score as the detection score.

### 4.2. Challenge I: Estimating Training Statistics at Scale

In OOD detection, many methods rely on statistics estimated from training features, such as the empirical mean or low-rank structure (e.g., SVD). For large language models, however, computing such statistics over the full training corpus is infeasible due to its sheer scale. Even estimating them from a sampled subset is challenging: the training data are extremely diverse, and a limited subset can easily introduce bias into the estimation. As a result, purely data-driven estimates of training statistics become neither practical nor reliable in this setting. This motivates the need for *analytical, model-intrinsic estimations* of training statistics that do not require access to the training data.

In the following, we take the estimation of the training feature mean $\mu_G$ in the **NCI** score (Definition 3.1) as a step toward bridging this gap. Since the language head $f_{\text{head}}$ is trained against the penultimate representations $z$, its parameters implicitly reflect the geometry of the training feature distribution. Specifically, the geometry of well-trained classifiers underlying NCI suggests that $\boldsymbol{\mu}_G$ lies near the maximal-uncertainty point. We thus adopt this point as an analytical proxy for $\boldsymbol{\mu}_G$ in the computation of NCI. Formally defining the proxy point as the *Decision-Neutral Closest Point*, obtained by minimizing the variance of logits across the vocabulary.

**Lemma 4.1** (Analytical Solution for Decision-Neutral Closest Point). *Let $W \in \mathbb{R}^{|\mathcal{V}| \times d_{model}}$ be the weight matrix whose*

*Table 1.* **Analytical Proxy of Feature Mean Enables Effect Hallucination Detection.** NCI with the analytical proxy outperforms the empirical estimation variant and Perplexity. AUROC reported for CSQA on `Llama-3.2-3B-Instruct`, with higher values indicating better performance; the best result is shown in **bold**.

| Method | AUROC |
|---|---|
| Perplexity | 63.23 |
| NCI w/ empirical estimation | 62.79 |
| NCI w/ analytical proxy | **66.07** |

*Table 2.* **Controlling the alternative set size $k$ improves performance of fDBD.** AUROC on CSQA using `Llama-3.2-3B-Instruct` (higher is better). The best-performing $k$ and the corresponding AUROC are highlighted in **bold**. While the method is effective across all choices of $k$, peak performance is achieved at $k = 1\,000$.

| $k$ | AUROC |
|---|---|
| 1 | 68.64 |
| 10 | 68.76 |
| 100 | 69.18 |
| **1 000** | **69.24** |
| 10 000 | 68.87 |
| 100 000 | 68.26 |
| All | 68.15 |

rows are $\boldsymbol{w}_v^\top$, and let $\boldsymbol{b} \in \mathbb{R}^{|\mathcal{V}|}$ be the vector of biases $b_v$. The point $\boldsymbol{z}^\star$ that minimizes logit variance across $\mathcal{V}$ is given by:

$$\hat{\boldsymbol{z}}_\star = -(W^\top P W)^\dagger W^\top P \boldsymbol{b} \tag{4}$$

where $P = I - \frac{1}{|\mathcal{V}|}\boldsymbol{1}\boldsymbol{1}^\top$ with $I$ denoting the identity matrix, $\boldsymbol{1}$ the all-ones vector, and $(\cdot)^\dagger$ the Moore–Penrose pseudo-inverse.

The formal definition of the optimization objective of Decision-Neutral Point and the detailed derivation are provided in Appendix J.

We empirically evaluate the effectiveness of this analytical proxy for hallucination detection on CSQA using `Llama-3.2-3B-Instruct` (Table 1). Notably, the model employs a zero-bias language head ($\boldsymbol{b} = \boldsymbol{0} \in \mathbb{R}^{|\mathcal{V}|}$), implying that the analytical estimate of the mean by Equation 4 reduces to the origin of the feature space, $\boldsymbol{0} \in \mathbb{R}^{d_{\mathrm{model}}}$. For comparison, we also consider an empirical estimate of the training feature mean computed from the CSQA training set. Both the analytical proxy and the empirical estimate are incorporated into the step-wise NCI score (Equation 2). The final hallucination decision obtained by thresholding the average score across an output sequence of length $T$, i.e.,

$$S_{\mathrm{NCI}} = \frac{1}{T}\sum_{t=0}^{T-1} s_{\mathrm{NCI}}(\boldsymbol{z}^t).$$

We further compare these variants against the Perplexity baseline, with results reported in Table 1.

As shown in Table 1, incorporating the analytical proxy of $\boldsymbol{\mu}_G$ into the NCI score enables effective hallucination detection, outperforming both the baseline Perplexity and the NCI variant using the empirical mean. This demonstrates the effectiveness of analytically estimating training statistics for hallucination detection.

### 4.3. Challenge II: Effectiveness and Efficiency in Massive Vocabulary Space

Another challenge in extending OOD detectors to hallucination detection lies in the dramatically increased label space. While conventional classifiers typically involve on the order

of thousands of classes at most, large language models operate over vocabularies containing hundreds of thousands of tokens. This scale poses challenges for directly applying OOD detectors such as **fDBD**, which indiscriminately aggregates signals from the entire label space.

Specifically, a direct adaptation of the **fDBD** detector to hallucination detection would compute the normalized average distance to the decision boundary over all tokens in the vocabulary:

$$s_{\mathrm{fDBD}} := \frac{1}{|\mathcal{V}| - 1} \sum_{c \in \mathcal{V}, c \neq \hat{c}} \frac{\tilde{D}_f(\boldsymbol{z}^t, c)}{\|\boldsymbol{z}^t - \boldsymbol{\mu}_G\|_2}, \tag{5}$$

where $\tilde{D}_f(\boldsymbol{z}^t, c)$ is as defined in Definition 3.2. This formulation raises both effectiveness and efficiency concerns in the context of large vocabularies.

Intuitively, distances to the boundaries of low-probability tokens—such as irrelevant symbols, rare tokens, or numbers during plain-text reasoning—tend to be consistently large. These distant boundaries may offer less informative uncertainty signals and risk diluting the more meaningful distances associated with semantically plausible alternative tokens. Furthermore, Equation 5 scales linearly with vocabulary size, taking $O(d_{model}|\mathcal{V}|)$ computes per step; while feasible, such overhead is less than ideal.

Motivated by these, we restrict the distance computation to a small set of confident alternative tokens. Specifically, we select $k$ tokens with the highest logits at step $t$, excluding the top-ranked token $\hat{c}$ (which is omitted as the feature $\boldsymbol{z}^t$ already resides within its decision region). And we compute the average normalized distance to the decision boundaries only with in the top-k set $\mathcal{K}_t$:

$$s_{\mathrm{fDBD}}^k := \frac{1}{k} \sum_{c \in \mathcal{K}_t} \frac{\tilde{D}_f(\boldsymbol{z}^t, c)}{\|\boldsymbol{z}^t - \boldsymbol{\mu}_G\|_2}. \tag{6}$$

This discards the potentially noisy uncertainty scores from

*Table 3.* **OOD-inspired hallucination detectors, NCI and fDBD, retain strong performance under stochastic decoding**, significantly outperforming the baseline Perplexity. Results on CSQA using `Llama-3.2-3B-Instruct`. AUROC is reported (higher is better). For each temperature, mean $\pm$ standard deviation is computed over five random seeds.

| Method | temp = 0.2 | temp = 0.5 | temp = 0.8 | temp = 1.0 |
|---|---|---|---|---|
| Perplexity | $63.49 \pm 0.75$ | $62.08 \pm 2.34$ | $63.45 \pm 0.71$ | $62.68 \pm 1.08$ |
| NCI | $67.07 \pm 0.53$ | $66.04 \pm 1.59$ | $67.53 \pm 0.88$ | $67.93 \pm 1.65$ |
| fDBD | $69.30 \pm 0.56$ | $68.19 \pm 1.47$ | $69.12 \pm 0.56$ | $69.19 \pm 1.98$ |

low-probability tokens, while reducing the expected complexity at each step from $O(d_{model}|\mathcal{V}|)$ to $O(d_{model}k + |\mathcal{V}|)$ [1]. In practice, $k$ can be chosen from the validation set as discussed in Section 5.

Empirically, controlling alternative set size gives benefit, as shown with CSQA on `Llama-3.2-3B` in Table 2 In these experiments, we use the analytical estimate $z^\star$ for $\mu_G$ as described in Section 4.2 and detect hallucination by thresholding the average score across $T$ decoding steps for the output sequence of length $T$,

$$S_{\text{fDBD}}^k = \frac{1}{T} \sum_{t=1}^{T} s_{\text{fDBD}}^k(z^t), \qquad (7)$$

Specifically, we consider $k = $ All, which includes all alternative tokens, as well as $k \in \{1, 10, 100, 1\,000, 10\,000, 100\,000\}$. As measured by AUROC (higher is better), the method remains effective across all choices of $k$ consistently outperforming the Perplexity baseline (AUROC = 63.23). Performance peaks at $k = 1\,000$, suggesting that appropriately controlling the size of the alternative token set yields additional performance gains.

### 4.4. Challenge III: Robustness to Stochastic Generation

While conventional classifiers produce deterministic predictions, large language models can generate outputs either deterministically (e.g., via greedy decoding) or stochastically. So far, our experiments focus on greedy decoding, aligning with the deterministic setting of conventional classification problems. Under stochastic decoding, however, the generated token at each step may differ from the token with the highest logit. This poses a potential challenge for our hallucination detectors **NCI** and **fDBD**, as both are adapted from OOD detectors and are defined with respect to the highest-logit token. Specifically, NCI measures the proximity of the hidden representation to the weight vector corresponding to the highest logit, while fDBD treats the representation as residing within the decision region of that token of max logit—a notion that may become less precise under stochastic decoding.

However, we hypothesize that both metrics would "catch up" over the course of a reasoning task and retain strong performance on a sequence level. The geometric uncertainty measures in NCI and fDBD capture the model's internal uncertainty at each decoding state. When the generated token aligns with this internal uncertainty—low-probability tokens in uncertain states or high-probability tokens in certain states—the measures accurately reflect per-step generation confidence. Misalignment can occur under stochastic decoding, for example, when sampling produces a low-probability token in an otherwise certain state (the converse case is practically infeasible). Such mismatches are typically short-lived: the model tends to transition into genuinely uncertain states in subsequent steps, allowing the uncertainty signal to quickly recover and remain reliable over the remainder of the sequence.

Table 3 validates our hypothesis and shows strong performance of NCI and fDBD under stochastic decoding. We evaluate performance across a range of standard decoding temperatures $temp \in \{0.2, 0.5, 0.8, 1.0\}$. For each temperature setting, we conduct five independent runs using different random seeds and report the mean AUROC along with its standard deviation. For simplicity, we set alternative set size $k = $ All. As observed in the table, NCI and fDBD maintain strong performance across all temperatures and consistently outperform the baseline Perplexity.

We further validate our hypothesis in Appendix K by considering a decoded-token-based framework as an alternative. While fDBD does not readily extend to this setting, NCI adapts naturally by measuring feature proximity to the decoded token's weight vector. Empirically, this variant achieves performance comparable to the original NCI, confirming that imprecise alignment during stochastic decoding does not degrade sequence-level performance.

## 5. Experiments

In this section, we evaluate the hallucination detectors NCI and fDBD across additional datasets and model architectures to assess their generalization. We compare our methods against a broader set of baselines, where NCI and fDBD consistently achieve superior performance. We further analyze the sensitivity of the hyperparameter $k$, which controls the size of the alternative token set in fDBD.

---

[1] `Quickselect`(Hoare, 1961) enables selecting the top-$k$ tokens (unordered) in $O(|\mathcal{V}|)$ expected time.

*Table 4.* **OOD-inspired hallucination detectors NCI and fDBD achieve superior performance with minimal latency overhead.**

*(a)* **OOD-inspired hallucination detectors NCI and fDBD achieve superior performance across different datasets and model architectures.** Our methods, NCI, fDBD with $k$ set to ALL, and fDBD with $k$ selected on the validation set, are highlighted in shading. "Single Sample" indicates whether the method requires multiple samples for hallucination detection. All methods are *training-free*. Performance is reported in AUROC (higher is better). Best performance is shown in **bold**, second best is underlined.

| Model | Method | Single Sample | CSQA | GSM8K | AQuA |
|---|---|:---:|:---:|:---:|:---:|
| Llama-3.2-3B-Instruct | Perplexity | ✓ | 63.23 | 69.63 | 72.85 |
| | Predictive Probability | ✓ | 61.63 | 70.88 | 69.07 |
| | LN Predictive Probability | ✓ | 61.51 | 70.68 | 68.98 |
| | Max P | ✓ | 66.01 | 73.90 | 66.02 |
| | P(True) | ✓ | 47.73 | 51.02 | 39.38 |
| | CoE-R | ✓ | 47.06 | 50.12 | 45.55 |
| | CoE-C | ✓ | 58.82 | 60.69 | 62.56 |
| | Lexical Similarity | ✗ | 62.94 | 73.66 | 71.48 |
| | SelfCheckGPT NLI | ✗ | 64.18 | 74.29 | 66.01 |
| | Semantic Entropy | ✗ | 60.61 | 64.40 | 64.71 |
| | **NCI** | ✓ | 66.07 | 76.32 | 74.41 |
| | **fDBD** | ✓ | 68.15 | 75.59 | 75.80 |
| | **fDBD (selected k)** | ✓ | **69.24** | **76.36** | **76.20** |
| Qwen-2.5-7B-Instruct | Perplexity | ✓ | 61.94 | 71.54 | 71.66 |
| | Predictive Probability | ✓ | 64.91 | 73.29 | 73.37 |
| | LN Predictive Probability | ✓ | 65.19 | 73.01 | 74.17 |
| | Max P | ✓ | 49.90 | 50.00 | 50.83 |
| | P(True) | ✓ | 68.01 | 70.31 | 72.86 |
| | CoE-R | ✓ | 62.75 | 75.13 | 72.13 |
| | CoE-C | ✓ | 66.89 | 75.50 | 72.04 |
| | Lexical Similarity | ✗ | 60.57 | 72.02 | 72.62 |
| | SelfCheckGPT NLI | ✗ | 60.18 | 76.22 | 70.90 |
| | Semantic Entropy | ✗ | 59.10 | 66.83 | 69.62 |
| | **NCI** | ✓ | 71.60 | 75.83 | 78.19 |
| | **fDBD** | ✓ | 71.50 | 77.19 | 77.08 |
| | **fDBD (selected k)** | ✓ | **72.47** | **77.19** | **78.22** |

*(b)* **OOD-inspired hallucination detectors NCI and fDBD incur minimal latency overhead.** End-to-end inference latency (ms/token) on CSQA with `Llama-3.2-3B-Instruct` ($k = 1000$ for fDBD, as selected in Table 4a). Lower is better.

| | Standard | Perplexity | NCI | fDBD |
|---|:---:|:---:|:---:|:---:|
| Latency (ms/token) | 31.94 | 32.88 | 32.54 | 32.71 |

## 5.1. Main Results

**Datasets** In addition to the commonsense reasoning task CSQA, we evaluate on two mathematical reasoning benchmarks: GSM8K (Cobbe et al., 2021), which requires free-form numerical answers, and AQuA (Ling et al., 2017), which uses a multiple-choice format (see Appendix G for examples). We evaluate on the CSQA validation set (1,221 questions), the GSM8K test set (1,319 questions), and the AQuA validation split (254 questions). For each dataset, hyperparameter $k$, which controls the alternative set size, is selected with training subset.

**Models** We extend the analysis from `Llama-3.2-3B-Instruct` (Section 4) to `Qwen-2.5-7B-Instruct` (Qwen et al., 2025) to assess generality across model families. In addition, we provide extensive validation of our methods in the appendices, including scalability to larger models with `Qwen-3-32B` (Yang et al., 2025) (Appendix A), generalizability across model sizes within the same family (Appendix B), generalizability to non-instruction-tuned base models (Appendix C), generalizability to MoE models (Appendix D), and generalizability to alternative architectures (Appendix E).

*Table 5.* **OOD inspired hallucination detector fDBD exhibits low sensitivity to the choice of $k$, which controls the size of the alternative token set.** AUROC is reported. Experiments are conducted on `Llama-3.2-3B-Instruct`. The selected values of $k$ are 1 000 for CSQA and 100 for GSM8K and AQuA.

| Dataset | Variation in $k$ | | | | | | |
|---|---|---|---|---|---|---|---|
| | $-20\%$ | $-10\%$ | $-5\%$ | **Selected** $(k)$ | $+5\%$ | $+10\%$ | $+20\%$ |
| CSQA | 69.25 | 69.24 | 69.24 | **69.24** | 69.23 | 69.23 | 69.22 |
| GSM8K | 76.41 | 76.38 | 76.36 | **76.36** | 76.34 | 76.34 | 76.32 |
| AQuA | 76.34 | 76.26 | 76.26 | **76.20** | 76.21 | 76.23 | 76.15 |

**Baselines** We compare our method against *training-free* hallucination detectors that operate on a *single sample*. In addition to Perplexity (Section 4), we consider common baselines: Predictive Probability, i.e., $\prod_{t=1}^{T} p(\boldsymbol{y}_t \mid \boldsymbol{x}, \boldsymbol{y}_{<t})$, Length-Normalized (LN) Predictive Probability, which normalizes predictive probability by output sequence length; Perplexity (Si et al., 2022), which measures model predictive uncertainty; Maximum Softmax Probability (Max P) (Wang et al., 2024); and verbalized method P(True) (Kadavath et al., 2022), which queries model confidence through an auxiliary prompt verbatim, as in the original paper. We also consider latent-space Chain-of-Embeddings (CoE) methods (Wang et al., 2024), including CoE-R and CoE-C.

In addition, we compare against baselines that detect hallucinations by measuring consistency across *multiple samples*. Specifically, we compare against Lexical Similarity (Lin et al., 2022b; 2023), which measures consistency using ROUGE-L scores; SelfCheckGPT-NLI (Manakul et al., 2023), which measures discrepancies via contradiction scores from natural language inference; and Semantic Entropy (Kuhn et al., 2023b), which clusters similar texts and then computes entropy across clusters. In our experiments, we set the temperature to $0.5$ and the sample size to three. We consider different sample size in Appendix F.

**Effectiveness** Along with the baselines, we evaluate our hallucination detectors NCI and fDBD. All experiments are conducted under greedy decoding ($T = 0$). AUROC is reported, where higher values indicate better performance. We note that AUROC is conventionally reported in the range $[50, 100]$; following prior work (Wang et al., 2024), we allow values outside this range when hallucinated samples receive higher confidence scores than non-hallucinated ones, to facilitate comparison across methods.

For fDBD, we consider two variants: the hyperparameter-free version that considers distances to all decision boundaries, and a version that computes distances only to the top $k$ candidate tokens, with $k \in \{1, 10, 100, 1\,000, 10\,000, 100\,000, \text{All}\}$. Based on validation results, we select $k = 1\,000, 100, 100$ on `Llama-3.2-3B-Instruct` for CSQA, GSM8K, and AQuA respectively; and $k = 100, \text{All}, 100$ on

`Qwen-2.5-7B-Instruct` for CSQA, GSM8K, and AQuA respectively.

Examining the results, fDBD with a selected subset of size $k$ almost always outperforms the full fDBD variant, confirming the benefit of controlling the size of the alternative token set. In most cases, fDBD also outperforms NCI. This reflects the efficiency–effectiveness trade-off between the two methods: NCI has per-step complexity $O(d_{\text{model}})$, making it computationally cheaper than fDBD, which has complexity $O(d_{\text{model}}|\mathcal{V}|)$ in its full form and $O(d_{\text{model}}k+|\mathcal{V}|)$ when restricted to a selected subset of size $k$. Both NCI and fDBD consistently outperform single-sample baselines.

We further observe that multiple-sampling based methods—Lexical Similarity, SelfCheckGPT NLI, and Semantic Entropy—are less effective on reasoning tasks compared to their original scope of concise question-answering. This supports our intuition that measuring discrepancies across samples is intrinsically challenging in the presence of diverse reasoning chains. In contrast, our methods consistently surpass these approaches in the vast majority of cases, demonstrating strong reliability and computational efficiency across models.

**Inference Latency** In addition to the computational complexity analysis, we measure end-to-end inference latency on an A100 GPU (ms/token) for standard decoding (without hallucination detection), perplexity, and our detectors in Table 4b. Experiments are conducted on CSQA using `Llama-3.2-3B-Instruct`, with $k = 1000$ as selected in Table 4a. NCI and fDBD incur minimal latency overhead on top of standard decoding, comparable to perplexity.

## 5.2. Hyper-parameter Sensitivity

We next examine the sensitivity of fDBD performance to the choice of $k$, which controls the size of the alternative token set used for computing the decision boundary distance. To this end, we analyze variations of $k$ around the selected value for each dataset on *Llama-3.2-3B-Instruct*, where the selected $k$ is 1 000 for CSQA and 100 for GSM8K and AQuA. Specifically, we evaluate $k$ values obtained by perturbing the selected value by $\pm 5\%$, $\pm 10\%$, and $\pm 20\%$.

*Table 6.* **OOD-inspired hallucination detectors NCI and fDBD achieve strong performance on open-ended tasks.** Experiments on TruthfulQA with `Llama-3.2-3B-Instruct`. Performance is reported in AUROC (higher is better). Best performance is shown in **bold**, second best is underlined.

| Method | AUROC |
| --- | --- |
| Perplexity | 63.78 |
| Predictive Probability | 51.32 |
| LN Predictive Probability | 50.38 |
| Max P | 60.62 |
| CoE-R | 48.91 |
| CoE-C | 55.16 |
| **fDBD** | 67.83 |
| **NCI** | **68.17** |

As shown in Table 5, fDBD performance remains highly stable across all tested variations, with AUROC changing only marginally as $k$ varies. These results validate that fDBD is not overly sensitive to the exact choice of $k$. Selecting the order of magnitude of $k$, as done in Section 5.1, is sufficient, making it possible to transfer a value of $k$ selected on the validation set to the test set.

### 5.3. Generalizability to Open-ended Tasks

So far, we have focused on reasoning tasks. To evaluate generalizability to open-ended settings, we additionally experiment on TruthfulQA dataset (Lin et al., 2022a), a benchmark designed to test whether models generate truthful answers to questions where false but plausible misconceptions are common. We report AUROC on `Llama-3.2-3B-Instruct` in Table 6. For fDBD, we report results with $k = $ ALL. As shown, fDBD and NCI maintain competitive performance compared to single-sample baselines, validating their generalizability to open-ended tasks. See experimental details in Appendix H.

## 6. Related Work

### 6.1. Out-of-Distribution Detection

In the context of classifiers, extensive prior work has investigated out-of-distribution (OOD) detection, with the goal of identifying test samples whose classes were unseen during training. In particular, one line of work builds on pre-trained classifiers to design post-hoc OOD detectors. For example, (Hendrycks et al., 2019; Liang et al., 2018; Liu et al., 2020; Sun et al., 2021; Sun & Li, 2022; Xu et al., 2024) define OOD scores in the output space of a classifier, whereas (Lee et al., 2018; Sun et al., 2022; Liu & Qin, 2024; 2025) measure OOD-ness in the feature space. In this work, we conceptually bridge OOD detection in classifiers to hallucination detection in LLMs. Specifically, we

extend prior work on OOD detection, NCI(Liu & Qin, 2025) and fDBD(Liu & Qin, 2024), to the task of hallucination detection and demonstrate its strong performance in this setting.

### 6.2. Hallucination Detection

Hallucinations pose risks to the safe deployment of LLMs, and much existing research has therefore focused on detecting them at inference time. One line of work detects hallucinations by measuring consistency across multiple samples (Lin et al., 2023; 2022b; Manakul et al., 2023; Xiao & Wang, 2021; Kuhn et al., 2023a; Chen et al., 2024; Liu et al., 2026; Hou et al., 2024; Jiang et al., 2023; Gao et al., 2024). While effective, such methods introduce nontrivial inference overhead due to the need for repeated sampling. Another line of work aims to detect hallucinations from a single inference without repetitive sampling (Azaria & Mitchell, 2023; Kossen et al., 2024; Li et al., 2023a; Liu et al.; Manakul et al., 2023; Marks & Tegmark; Su et al., 2024; Wang et al., 2024). These methods usually require training a separate model, adding additional computational cost and suffering from training-test distribution shift. To address the limitations of both lines of work, we detect hallucinations by analyzing the distance between internal features and the model's decision boundary, yielding an inference-efficient and distribution-shift–robust approach.

**Limitations.** Our methods generally achieve strong performance on reasoning and open-ended tasks. However, their advantage can diminish on datasets such as TriviaQA (Joshi et al., 2017), where both correct and incorrect answers often span only a few tokens. Better understanding and mitigating this gap across answer formats is an important direction for future work. Our study is also limited to two geometric OOD scores. Exploring the broader literature on state-of-the-art OOD detection methods and adapting them to hallucination detection remains an open problem.

## 7. Conclusion

This work tackles hallucination detection in LLMs. We adapt existing OOD detectors to account for the structural differences in large language models, yielding training-free, single-sample detectors that effectively identify hallucinations in reasoning tasks. We hope this framework inspires further OOD-based approaches to enhance LLM safety.

## Impact Statement

This paper presents work whose goal is to advance the field of Machine Learning. There are many potential societal consequences of our work, none which we feel must be specifically highlighted here.

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

# A. Scalability to Larger Language Model

To evaluate the scalability of our approach to larger language models, we conduct experiments on `Qwen-3-32B` (Yang et al., 2025). For computational concerns, we focus on three computationally efficient baselines: Perplexity, Predictive Probability, and Length-Normalized (LN) Predictive Probability. For fDBD, we did not do hyperparameter sweeping of k and report with all k. As shown in Table 7, NCI achieves strong overall performance, while fDBD consistently outperforms these baselines across CSQA, GSM8K, and AQuA. These results demonstrate that methods adapted from OOD detection scale favorably to large language models.

*Table 7.* **OOD-inspired hallucination detectors scale to larger language models.** Results on `Qwen-3-32B`. We report AUROC (higher is better), with the best performance highlighted in **bold**. NCI achieves strong overall performance, while fDBD consistently outperforms the baselines across datasets.

| Method | CSQA | GSM8K | AQuA |
|---|---|---|---|
| Perplexity | 58.44 | 76.86 | 53.89 |
| Predictive Probability | 59.30 | 74.99 | 57.97 |
| LN Predictive Probability | 59.35 | 74.38 | 58.21 |
| **NCI** | 64.54 | 75.75 | 60.28 |
| **fDBD** | **65.68** | **80.60** | **65.37** |

# B. Generalizability Across Model Scales

We further evaluate hallucination detection performance across the Qwen3 family (0.6B, 1.7B, 4B, and 8B). We report AUROC on the CSQA dataset in Table 8. For fDBD we report with $k = $ ALL. Compared to single-sample baselines, NCI achieves strong overall performance across model sizes, whereas fDBD consistently achieves the strongest results among all methods.

*Table 8.* **OOD-inspired hallucination detectors achieve superior performance across Qwen3 models of different scales.** Results on CSQA dataset. We report AUROC (higher is better), with the best performance highlighted in **bold**. NCI achieves strong overall performance, while fDBD consistently outperforms the baselines across datasets.

| Method | Qwen-3-0.6B | Qwen-3-1.7B | Qwen-3-4B | Qwen-3-8B |
|---|---|---|---|---|
| Perplexity | 54.62 | 57.77 | 57.30 | 58.70 |
| Predictive Probability | 54.06 | 58.89 | 59.18 | 60.33 |
| LN Predictive Probability | 53.92 | 58.94 | 60.21 | 60.56 |
| Max P | 56.74 | 50.00 | 50.00 | 50.00 |
| CoE-R | 50.02 | 52.00 | 53.43 | 54.81 |
| CoE-C | 50.10 | 52.43 | 53.36 | 55.08 |
| **NCI** | 57.76 | 58.87 | 61.92 | 66.69 |
| **fDBD** | **58.94** | **59.59** | **63.90** | **66.93** |

# C. Generalizability to Base Models

Extending beyond the instruction-tuned models studied in the main paper, we validate the generalizability of our methods on base models. Specifically, we evaluate on CSQA with `Llama-3.2-3B-Base`. As shown in Table 9, our detectors outperform the baselines in AUROC, validating their generalizability to base models.

# D. Generalizability to MoE Models

Our methods remain applicable to Mixture-of-Experts (MoE) models, since their language modeling head remains dense; the MoE mechanism operates only within the transformer blocks. To empirically validate this, we evaluate our detectors on the AQuA dataset using the MoE model `Qwen-3-30B-A3B`. Table 10 compares our methods against single-sample detectors. For fDBD, we report results with $k = $ ALL. Compared to single-sample baselines, NCI achieves strong overall performance across model sizes, whereas fDBD consistently achieves the strongest results among all methods.

*Table 9.* **OOD-inspired hallucination detectors achieve superior performance on base models.** Results on CSQA dataset with `Llama-3.2-3B-Base`. We report AUROC (higher is better), with the best performance highlighted in **bold**.

| Method | AUROC |
|---|---|
| Perplexity | 59.09 |
| Predictive Probability | 60.44 |
| LN Predictive Probability | 60.43 |
| Max P | 55.51 |
| CoE-R | 47.56 |
| CoE-C | 51.21 |
| **NCI** | 62.65 |
| **fDBD** | **62.70** |

*Table 10.* **OOD-inspired hallucination detectors achieve superior performance on MoE models.** Results on AQuA dataset with MoE model `Qwen-3-30B-A3B`. We report AUROC (higher is better), with the best performance highlighted in **bold**.

| Method | AUROC |
|---|---|
| Perplexity | 65.40 |
| Predictive Probability | 52.82 |
| LN Predictive Probability | 52.04 |
| Max P | 50.00 |
| CoE-R | 49.22 |
| CoE-C | 58.98 |
| **NCI** | **70.09** |
| **fDBD** | 69.60 |

## E. Generalizability to Alternative Architectures

To test the applicability of our methods beyond Llama and Qwen models, we experiment with the alternative architecture `Mistral-7B-Instruct-v0.3` and report AUROC on CSQA in Table 11. Both fDBD and NCI maintain competitive performance compared to single-sample baselines, validating the generalizability of our methods to alternative architectures.

*Table 11.* **OOD-inspired hallucination detectors maintain superior performance on Mistral-7B-Instruct-v0.3, validating generalizability to alternative architectures.** We report AUROC (higher is better) on CSQA dataset. The best performance highlighted in **bold**.

| Method | AUROC |
|---|---|
| Perplexity | 58.96 |
| Predictive Probability | 58.82 |
| LN Predictive Probability | 58.75 |
| Max P | 57.75 |
| CoE-R | 49.82 |
| CoE-C | 62.34 |
| **NCI** | 64.91 |
| **fDBD** | **66.36** |

## F. Effect of Sample Size on Multi-sample Baselines

In Section 5.1, we evaluate multi-sample baselines with a fixed sample size of 3. To further examine whether increasing the number of samples improves relative performance, we evaluate SelfCheckGPT-NLI on AQuA with `Llama-3.2-3B-Instruct` using different sample sizes. As shown in Table 12, increasing the sample size does not substantially improve performance in this setting, and SelfCheckGPT-NLI remains below our methods. This provides additional evidence that multi-sample detectors may face challenges on reasoning tasks, where the inherent diversity of valid reasoning paths can make response-level comparisons difficult.

*Table 12.* **OOD-inspired hallucination detectors outperform baseline multi-sample detect across different sample size.** We report AUROC (higher is better) on AQuA dataset with `Llama-3.2-3B-Instruct`. The best performance highlighted in **bold**.

| Method | AUROC |
|---|---|
| SelfCheckGPT-NLI (sample size = 3) | 66.01 |
| SelfCheckGPT-NLI (sample size = 5) | 65.51 |
| SelfCheckGPT-NLI (sample size = 10) | 66.52 |
| **NCI** | 74.41 |
| **fDBD** | **75.80** |

## G. Implementation details for Reasoning Tasks

### G.1. Examples of Questions and Model Responses

#### G.1.1. CSQA EXAMPLE

**Question:** A revolving door is convenient for two direction travel, but it also serves as a security measure at a what?

(a) bank

(b) library

(c) department store

(d) mall

(e) new york

**Model response:** The answer should be a place where security is a priority. Of the above choices, a bank is the most likely to use a revolving door as a security measure. So the answer is (a).

#### G.1.2. GSM8K EXAMPLE

**Question:** Janet's ducks lay 16 eggs per day. She eats three for breakfast every morning and bakes muffins for her friends every day with four. She sells the remainder at the farmers' market daily for $2 per fresh duck egg. How much in dollars does she make every day at the farmers' market?

**Model response:** Janet's ducks lay 16 eggs per day. She uses 3 for breakfast and 4 for baking, which totals 3 + 4 = 7 eggs used daily. The remaining eggs are 16 - 7 = 9. She sells these for $2 each, so 9 x 2 = 18. The answer is 18.

#### G.1.3. AQUA EXAMPLE

**Question:** A password needs to contain 2 letters and 3 numbers. How many different passwords are possible if repetition of letters and numbers is allowed??

(a) 676000

(b) 676

(c) 100

(d) 6760

(e) 25

**Model response:** There are 26 letters in the alphabet and 10 digits (0-9). Since repetition is allowed, there are 26 * 26 = 676 ways to choose the letters and 10 * 10 * 10 = 1000 ways to choose the numbers. The total number of different passwords is 676 * 1000 = 676000. The answer is (a).

## G.2. Chain-of-Thought Few-shot Prompting

Following Wei et al. (2022), we use chain-of-thought few-shot prompting to elicit reasoning in the model's responses. For all datasets, we adopt the canonical examples from Wei et al. (2022). These prompts guide the model to produce a formatted answer after completing the reasoning process. For answer extraction, we extract the letter following *"So the answer is"* for CSQA, the number following *"The answer is"* for GSM8K, and the answer letter following *"The answer is"* for AQuA. An answer is considered correct if the extracted answer matches the ground truth.

## G.3. System Prompt

Since few-shot prompting alone is insufficient for some models (e.g. `Qwen-2.5-7B-Instruct`) to follow the required formatting, we further reinforce the format with the following system prompts. The system prompts remain consistent across all models in our experiments.

### G.3.1. CSQA SYSTEM PROMPT

> You are a plain-text reasoning engine. You must strictly follow the reasoning format shown in the examples. Do not start new paragraphs or lines. End each answer with: 'So the answer is (<letter>).'

### G.3.2. GSM8K SYSTEM PROMPT

> You are a mathematical reasoning engine. You must strictly follow the reasoning format shown in the examples. Use plain ASCII text only. Do not start new paragraphs or lines. End each answer with: 'The answer is <number>.'

### G.3.3. AQuA SYSTEM PROMPT

> You are a mathematical reasoning engine. You must strictly follow the reasoning format shown in the examples. Use plain ASCII text only. Do not start new paragraphs or lines. End each answer with: 'The answer is (<letter>).'

# H. Implementation details for Open-ended Tasks (TruthfulQA)

## H.1. Prompt Details

We omit the system prompt in TruthfulQA experiments, as `Llama-3.2-3B-Instruct` reliably follows the required format under few-shot prompting without it.

Following Li et al. (2023b) and prior work, we adopt the same few-shot prompting setup for TruthfulQA generation, as shown below. The prompt remains unchanged, except that, to better align with the goal of hallucination detection, we omit the abstention example ("I have no comment").

> **Q:** What is human life expectancy in the United States?
>
> **A:** Human life expectancy in the United States is 78 years.
>
> **Q:** Who was president of the United States in 1955?
>
> **A:** Dwight D. Eisenhower was president of the United States in 1955.
>
> **Q:** Which party did he belong to?
>
> **A:** He belonged to the Republican Party.
>
> **Q:** How does a telescope work?
>
> **A:** Telescopes use lenses or mirrors to focus light and make objects appear closer.
>
> **Q:** Where were the 1992 Olympics held?

**A:** The 1992 Olympics were held in Barcelona, Spain.

## H.2. LLM as Judge

Since TruthfulQA is an open-ended task, evaluating the correctness of model outputs requires an LLM judge. While the original TruthfulQA paper employs a fine-tuned GPT-3 model as the LLM judge, the corresponding API has since been deprecated; we instead use an open-source Qwen model (`Qwen-2.5-7B-Instruct`) with the following prompt.

You are a factual answer judge. You will be given a question, a reference best answer, and a model's answer. Respond with exactly one word — yes if the model's answer is factually correct and consistent with the reference answer, or no if it is incorrect or misleading. Do not explain. Output only: yes or no.

## I. Proof of Theorem 3.3

*Proof.* Let $\hat{c} = \arg\max_{v \in \mathcal{V}} \boldsymbol{w}_v^\top \boldsymbol{z} + b_v$ denote the model's most confident internal prediction for feature $\boldsymbol{z}$. For any alternative token $c \in \mathcal{V}$ with $c \neq \hat{c}$, recall that the decision region of token $c$ is

$$\mathcal{R}_c = \left\{ \boldsymbol{z} \in \mathbb{R}^{d_{\text{model}}} : \boldsymbol{w}_c^\top \boldsymbol{z} + b_c > \boldsymbol{w}_v^\top \boldsymbol{z} + b_v, \ \forall v \neq c \right\}.$$

We further define a relaxed region

$$\mathcal{R}_c' = \left\{ \boldsymbol{z} \in \mathbb{R}^{d_{\text{model}}} : \boldsymbol{w}_c^\top \boldsymbol{z} + b_c > \boldsymbol{w}_{\hat{c}}^\top \boldsymbol{z} + b_{\hat{c}} \right\},$$

which only enforces that token $c$ surpasses the current most confident token $\hat{c}$. By construction, $\mathcal{R}_c \subseteq \mathcal{R}_c'$. Therefore,

$$D_f(\boldsymbol{z}, c) = \inf_{\boldsymbol{z}' \in \mathcal{R}_c} \|\boldsymbol{z}' - \boldsymbol{z}\|_2 \geq \inf_{\boldsymbol{z}' \in \mathcal{R}_c'} \|\boldsymbol{z}' - \boldsymbol{z}\|_2. \tag{8}$$

Geometrically, the quantity $\inf_{\boldsymbol{z}' \in \mathcal{R}_c'} \|\boldsymbol{z}' - \boldsymbol{z}\|_2$ corresponds to the Euclidean distance from $\boldsymbol{z}$ to the hyperplane

$$(\boldsymbol{w}_{\hat{c}} - \boldsymbol{w}_c)^\top \boldsymbol{z} + (b_{\hat{c}} - b_c) = 0. \tag{9}$$

Thus,

$$\inf_{\boldsymbol{z}' \in \mathcal{R}_c'} \|\boldsymbol{z}' - \boldsymbol{z}\|_2 = \frac{\left| (\boldsymbol{w}_{\hat{c}} - \boldsymbol{w}_c)^\top \boldsymbol{z} + (b_{\hat{c}} - b_c) \right|}{\|\boldsymbol{w}_{\hat{c}} - \boldsymbol{w}_c\|_2}. \tag{10}$$

Combining (8) and (10) yields the lower bound in Eqn. (3).

Finally, let

$$c_2 = \arg \min_{c \in \mathcal{V}, \, c \neq \hat{c}} \inf_{\boldsymbol{z}' \in \mathcal{R}_c'} \|\boldsymbol{z}' - \boldsymbol{z}\|_2 \tag{11}$$

denote the token whose associated hyperplane is closest to $\boldsymbol{z}$, and let $\boldsymbol{p}$ be the projection of $\boldsymbol{z}$ onto this hyperplane. Since all other hyperplanes are farther from $\boldsymbol{z}$ than $\boldsymbol{p}$, the points $\boldsymbol{z}$ and $\boldsymbol{p}$ lie on the same side of every such hyperplane. Consequently, $\boldsymbol{p}$ lies in the closure of $\mathcal{R}_{c_2}$, implying

$$\|\boldsymbol{p} - \boldsymbol{z}\|_2 \geq \inf_{\boldsymbol{z}' \in \mathcal{R}_{c_2}} \|\boldsymbol{z}' - \boldsymbol{z}\|_2. \tag{12}$$

Together with the definition of $c_2$, this shows that the lower bound in Eqn. (3) is tight. $\square$

## J. Derivative of Decision-Neutral Point

Formally, for a language head, a decision-neutral point $z_\star$ is a feature at which all tokens receive identical logits, i.e.,

$$\boldsymbol{w}_v^\top \boldsymbol{z}_\star + b_v = \boldsymbol{w}_{v'}^\top \boldsymbol{z}_\star + b_{v'}, \forall v, v' \in \mathcal{V}$$

However, the resulting system of linear equations may be overdetermined and thus fail to admit a feasible solution. We therefore relax this notion and define the decision-neutral closest point.

**Definition J.1** (Decision-Neutral Closest Point). Given a classification head $f_{\text{head}}$, a *decision-neutral closest point* is the feature that minimizes the variance of logits across the entire vocabulary:

$$\hat{z}_\star = \arg\min_{z} \sum_{v \in \mathcal{V}} \left( w_v^\top z + b_v - \bar{l} \right)^2, \tag{13}$$

where $\bar{l} = \frac{1}{|\mathcal{V}|} \sum_{v \in \mathcal{V}} (w_v^\top z + b_v)$ represents the mean logit value across the vocabulary.

To derive analytical solution in Equation 4, we first convert the definition above to matrix form:

$$\hat{z}_\star = \arg\min_{z} \| P(Wz + b) \|^2 \tag{14}$$

where $P = I - \frac{1}{n} \mathbf{1}\mathbf{1}^\top$ is the symmetric and idempotent centering matrix. Define the function of $z$ as

$$f(z) = \| P(Wz + b) \|^2 \tag{15}$$

To find $\hat{z}_\star$, we now compute the gradient with respect to $z$:

$$\nabla_z f(z) = (PW)^\top \cdot 2[P(Wz + b)] \tag{16}$$
$$= 2W^\top P^\top P(Wz + b) \tag{17}$$

Setting the gradient to zero and we have:

$$\nabla_z f(z) = 2W^\top P(Wz + b) \tag{18}$$

The analytical solution is thus:

$$\hat{z}_\star = -(W^\top PW)^\dagger W^\top Pb, \tag{19}$$

where $(\cdot)^\dagger$ denotes the Moore–Penrose pseudo-inverse.

## K. Decoded-token based Variant of NCI

In addition to computing the NCI score with respect to the highest-logit token, we evaluate a variant where feature proximity is computed against the weight vector of the actually decoded token at each step. Under greedy decoding, this setup is the same as the adopted NCI in the main paper. Under stochastic decoding, this setup aligns the metric more directly with the stochastic output of the model. We evaluate performance across a range of standard decoding temperatures $temp \in \{0.2, 0.5, 0.8, 1.0\}$. For each temperature setting, we conduct five independent runs using different random seeds and report the mean AUROC along with its standard deviation. As shown in Table 13, the performance of this variant is similar to the max-logit version of NCI.

*Table 13.* **Two NCI variants, max logit token based and decoded token based, exhibit similar performance under stochastic decoding.** Results on CSQA using `Llama-3.2-3B-Instruct`. AUROC is reported (higher is better). For each temperature, mean $\pm$ standard deviation is computed over five random seeds.

| Method | temp = 0.2 | temp = 0.5 | temp = 0.8 | temp = 1.0 |
|---|---|---|---|---|
| NCI (max-logit token) | $67.07 \pm 0.53$ | $66.04 \pm 1.59$ | $67.53 \pm 0.88$ | $67.93 \pm 1.65$ |
| NCI (decoded token) | $67.10 \pm 0.52$ | $66.04 \pm 1.63$ | $67.44 \pm 0.86$ | $67.70 \pm 1.53$ |

