# OpenReview forum: "From Out-of-Distribution Detection to Hallucination Detection: A Geometric View"
_ICML.cc/2026/Conference — ICML 2026 regular_

### Official Review · Reviewer_CTX3 · 2026-03-04

**Soundness:** 3
**Presentation:** 2
**Significance:** 3
**Originality:** 2
**Overall Recommendation:** 4
**Confidence:** 3

**Summary:**

This paper reframes LLM hallucination detection as a geometric Out-of-Distribution (OOD) problem, proposing two training-free, single-sample detectors by adapt existing OOD detectors (NCI and fDBD), effectively identify hallucinations in reasoning task.

**Compliance With Llm Reviewing Policy:**

Affirmed.

**Final Justification:**

The authors have provided a wealth of empirical evidence to support the effectiveness of their method, I have decided on a final score of 4. However, they did not adequately explain why the theoretical assumptions hold under the specific pressure of preference-based optimization. Given the paper's emphasis on a "Geometric View," the lack of a clear analytical discussion on how modern alignment techniques reshape this geometry is a missing piece that prevents a higher score.

**Key Questions For Authors:**

Q1: The evaluation lacks a comparative analysis across different model sizes within the same family. The comparisons made across Qwen-2.5 variants (1.5B, 3B, 7B, 14B) or Qwen-3 variants (0.6B, 1.7B, 4B, 8B) should be provided.

Q2: The method's derivation relies on the assumption that the language head is linear, which holds for dense models (like Qwen-2.5 and Llama-3.2). Can this assumption adapt well to the dynamic routing mechanism in Mixture of Experts (MoE) models (like Qwen-3)?

Q3: Theoretically, the analysis is more applicable to Base models like Llama-3.2-3B-Base rather than instruction-tuned models.

* Q3.1: The experiments utilize Llama-3.2-3B-Instruct trained with RLHF or DPO, yet the paper does not analysis theoretically weather RLHF or DPO alignment distort the feature space geometry assumed by the method.
* Q3.2: The paper should provide a comparison between Llama-3.2-3B-Base and Llama-3.2-3B-Instruct to demonstrate the specific impact of alignment on the proposed method.

**Limitations:**

L1: The current evaluation is strictly focused on reasoning tasks with verifiable ground truth. The applicability of the method to open-ended generation tasks remains uncertain.

L2: The theoretical assumptions of the method may not adapt well to the most advanced LLM based on MoE architectures or those subjected to strong RLHF/DPO alignment.

**Strengths And Weaknesses:**

Strengths

S1: The adaptation of geometric OOD concepts to the LLM domain is innovative.

S2: The paper provides a lucid analysis of the specific challenges involved in applying traditional OOD detection to LLMs.

S3: The proposed training-free, single-sample detectors are extensively evaluated across multiple models and datasets, demonstrating robust performance in detecting hallucinations within reasoning tasks.

Weaknesses

W1: While the paper theoretically argues for low computational complexity, the proposed method necessitates additional geometric calculations at every generation step. The experiments should provide specific inference latency overhead (in ms/token) compared to standard decoding and other hallucination detectors.

W2: The theoretical assumptions of the paper are only reasonable in some LLM model architectures, but are not applicable in the most advanced LLM models. Please refer to Q2 and Q3 for details.

---

> ### Author Rebuttal · Authors · 2026-03-29
>
> Thank you for your helpful feedback. Please see our responses below. We will include the additional experiments in the revision.
>
> **W1: Inference Latency**
>
> Following your suggestion, we report inference latency on A100 (ms/token) for standard decoding, perplexity, and our detectors in Table 1. Experiments use CSQA with Llama-3.2-3B-Instruct with k=1000 as in Section 5.1. NCI and fDBD add minimal latency overhead, comparable to perplexity.
>
> | Method | Latency (ms/token) |
> |--------------|--------------------|
> | Standard | 31.94 |
> | Perplexity | 32.88 |
> | NCI | 32.54 |
> | fDBD | 32.71 |
>
> Table 1: NCI and fDBD add minimal latency overhead, comparable to perplexity.
>
>
> **W2 / Q2 / L2:  Applicability to MoE models**.
>
> Our assumption is applicable to MoE models, since their language head remains dense – MoE operates only within transformer blocks. In Table 2,  we empirically validate the effectiveness of our detectors on the MoE model Qwen-3-30B-A3B, comparing against single-sample detectors.
>
> |  Method | AUROC|
> |--------------------------|-------|
> | Predictive Probability | 52.82 |
> | LN Predictive Probability| 52.04 |
> | max(P) | 50.00 |
> | Perplexity | 65.40 |
> | COE_r | 49.22 |
> | COE_c | 58.98 |
> | NCI | 70.09 |
> | fDBD | 69.60 |
>
> Table 2: NCI and fDBD remain effective on Qwen-3-30B-A3B (MoE).
>
>
>
> **Q1: Models of the Same Family**
>
> Following your suggestion, we experiment across the Qwen3 family (0.6B, 1.7B, 4B, and 8B), reporting AUROC on CSQA in Table 3. fDBD and NCI maintain competitive performance across model sizes compared to single-sample baselines.
>
> | Method | Qwen-3-0.6B | Qwen-3-1.7B | Qwen-3-4B | Qwen-3-8B |
> |---------------------------|-------------|-------------|-----------|-----------|
> | Predictive Probability | 54.06 | 58.89 | 59.18 | 60.33 |
> | LN Predictive Probability | 53.92 | 58.94 | 60.21 | 60.56 |
> | max(P) | 56.74 | 50.00 | 50.00 | 50.00 |
> | Perplexity | 54.62 | 57.77 | 57.30 | 58.70 |
> | COE_r | 50.02 | 52.00 | 53.43 | 54.81 |
> | COE_c | 50.10 | 52.43 | 53.36 | 55.08 |
> | NCI | 57.76 | 58.87 | 61.92 | 66.69 |
> | fDBD | 58.94 | 59.59 | 63.90 | 66.93 |
>
> Table 3: NCI and fDBD maintain competitive performance across the Qwen3 family.
>
>
> **W2 / Q3 / L2:  Applicability after instruction fine-tuning**
>
> Thank you for the insightful question.
>
> We agree that aligning with our detectors, cross-entropy loss **explicitly** optimizes for high logits per token, which encourages proximity between features and weight vectors.
> For RLHF/DPO, the proximity is encouraged more **implicitly**: RLHF/DPO updates the model so as to increase the likelihood of high-reward responses. And the increased likelihood requires higher logits, and thus induces higher proximity between feature and the weight vectors in tokens of the high-reward responses.
>
> We validate this intuition in Table 4 by explicitly comparing  the performance of Llama-3.2-3B-Base against Llama-3.2-3B-Instruct. Our detectors consistently outperform the baselines in AUROC on CSQA, suggesting that they remain effective both before and after instruction fine-tuning.
>
> |  Method | Llama-3.2-3B-Base | Llama-3.2-3B-Instruct |
> |--------------------------|------------------|-----------------------|
> | Predictive Probability | 60.44 | 61.63 |
> | LN Predictive Probability| 60.43 | 61.51 |
> | max(P) | 55.51 | 66.01 |
> | Perplexity | 59.09 | 63.23 |
> | COE_r | 47.56 | 47.06 |
> | COE_c | 51.21 | 58.82 |
> | NCI | 62.65 | 66.07 |
> | fDBD | 62.70 | 68.15 |
>
> Table 4: NCI and fDBD are effective both before and after instruction fine-tuning.
>
>
> **L1: Generalizability to Open-ended Task**
>
> Following your suggestion, we experiment on the open-ended TruthfulQA dataset, reporting AUROC on Llama-3.2-3B-Instruct in Table 5. fDBD and NCI maintain competitive performance compared to single-sample baselines, validating generalizability to open-ended tasks.
>
>
> | Method | AUROC |
> |----------------------------|--------|
> | Predictive Probability | 51.32 |
> | LN Predictive Probability | 50.38 |
> | max(P) | 60.62 |
> | Perplexity | 63.78 |
> | COE_r | 48.91 |
> | COE_c | 55.16 |
> | fDBD | 67.83 |
> | NCI | 68.17 |
>
> Table 5: NCI and fDBD maintain competitive performance on TruthfulQA, validating generalizability to open-ended tasks.

---

> > ### Author Rebuttal · Reviewer_CTX3 · 2026-04-01
> >
> > I have read the authors' responses and appreciate the effort they put into addressing my concerns.
> >
> > Regarding Q3.1, I acknowledge the authors' explanation that RLHF/DPO implicitly brings features closer to target weight vectors to boost the likelihood of high-reward responses. Yet, this only explains the positive 'pull.' It fails to adequately address my main point: preference-based optimization explicitly penalizes (pushes away) bad responses. This inevitably pushes features away from incorrect weights, effectively forming a 'rejection region.' How does this affect the fDBD decision boundaries?
> >
> > The authors' extensive experiments thoroughly validate their approach, and I will increase my score (3->4). However, the issue raised above still requires further clarification or discussion. If the authors can provide a deeper discussion on this point, I would be open to raising my score even further.
> >
> > ---
> > The latest reply still does not directly address this question. The response remains largely empirical and does not yet provide a clear analysis of how the rejection-region effect induced by preference optimization affects the geometry of the fDBD decision boundaries. If this point cannot be clarified further, I would be more comfortable keeping my original score of 3.

---

> > > ### Author Response · Authors · 2026-04-02
> > >
> > > We appreciate your support and your insightful question. We agree that RLHF pushes features away from incorrect weight vectors. This results in an overall increase in distance to decision boundaries, as we empirically validate on CSQA (Table 1). Meanwhile, the increase is larger for non-hallucinated responses due to an additional pull toward their correct weight vectors. Thus, the relative structure of fDBD is preserved: hallucinated features remain closer to decision boundaries.
> > >
> > > |  | Hallucination | Non-Hallucination |
> > > |---------------------|--------------|-------------------|
> > > | Llama-3.2-3B-Base | 14.51 | 14.60 |
> > > | Llama-3.2-3B-Instruct | 14.92 | 15.23 |
> > > | Δ (Instruct − Base) | +0.41 | +0.63 |
> > >
> > > Table 1: RLHF/DPO increases distances to decision boundaries, with a larger effect for non-hallucinated responses.

---

### Official Review · Reviewer_6ayY · 2026-03-09

**Soundness:** 3
**Presentation:** 3
**Significance:** 3
**Originality:** 3
**Overall Recommendation:** 5
**Confidence:** 3

**Summary:**

This paper innovatively transfers well-established out-of-distribution (OOD) detection techniques to the hallucination detection task in LLM. The authors argue that both OOD detection and hallucination detection fundamentally measure model uncertainty, thereby establishing a conceptual connection between the two tasks. Based on this insight, the paper use two lightweight OOD detectors, NCI and fDBD, to identify hallucinations without additional training. Experimental results on reasoning benchmarks such as CSQA and GSM8K demonstrate the effectiveness of the proposed approach.

**Compliance With Llm Reviewing Policy:**

Affirmed.

**Final Justification:**

This work applies OOD detection techniques from the computer vision domain to the problem of hallucination detection, which demonstrates strong novelty compared with methods that extract hidden-layer features for binary classification. The proposed approach also achieves promising experimental results.
However, as other reviewers have pointed out, this paper relies heavily on empirical assumptions and lacks in-depth theoretical analysis. Addressing this issue would significantly improve the overall quality and rigor of the manuscript.

**Key Questions For Authors:**

no other questions

**Limitations:**

As noted in the weakness, the method can be tested on diverse datasets to evaluate its robustness.

**Strengths And Weaknesses:**

### Strengths

1. **Novel Perspective Bridging OOD and Hallucination Detection**
    The paper identifies an intrinsic connection between hallucination detection and OOD detection, and leverages mature OOD techniques to address hallucination detection in LLMs. This perspective provides a novel geometric interpretation of hallucination.

2. **Training-Free and Efficient Design**
    The proposed approach does not require additional training or multiple inference passes, making it highly practical and computationally efficient. This design significantly enhances its applicability in real-world deployment scenarios.

3. **Comprehensive Experimental Evaluation**
    The experiments are conducted on multiple models with different architectures and scales, including LLaMA and Qwen. The consistent improvements across models strengthen the credibility and robustness of the empirical findings.
### Weaknesses
The current experimental evaluation focuses primarily on reasoning-oriented datasets. To further validate the generalization ability of the proposed method, it would be beneficial to include experiments on more general-purpose or open-domain benchmarks.

---

> ### Author Rebuttal · Authors · 2026-03-29
>
> Thank you for supporting our paper and for your valuable feedback!
>
> Following your suggestion, we experimented on the open-ended TruthfulQA dataset, reporting AUROC on Llama-3.2-3B-Instruct in Table 1. fDBD and NCI maintain competitive performance compared to single-sample baselines, validating generalizability to open-ended tasks. We will include the experiments in our revision.
>
> | Method | AUROC |
> |----------------------------|--------|
> | Predictive Probability | 51.32 |
> | LN Predictive Probability | 50.38 |
> | max(P) | 60.62 |
> | Perplexity | 63.78 |
> | COE_r | 48.91 |
> | COE_c | 55.16 |
> | fDBD | 67.83 |
> | NCI | 68.17 |
>
> Table 1: NCI and fDBD maintain competitive performance on TruthfulQA, validating generalizability to open-ended tasks.

---

> > ### Author Rebuttal · Reviewer_6ayY · 2026-04-01
> >
> > Thank you for the reviewer's rebuttal, which has addressed my concerns regarding the paper.

---

> > > ### Author Response · Authors · 2026-04-06
> > >
> > > We are pleased to learn that your concerns have been resolved. Thank you again for your constructive feedback.

---

### Official Review · Reviewer_2t8v · 2026-03-11

**Soundness:** 3
**Presentation:** 3
**Significance:** 4
**Originality:** 3
**Overall Recommendation:** 4
**Confidence:** 3

**Summary:**

This paper reframes hallucination detection in LLMs as an OOD-style uncertainty estimation problem and adapts two geometric detectors, NCI and fDBD, to token generation through the language head’s geometry. It introduces an analytical proxy for the training feature mean and a top-k variant of fDBD to make boundary-based scoring practical in very large vocabularies.

**Compliance With Llm Reviewing Policy:**

Affirmed.

**Final Justification:**

Since all my concerns have been resolved, I recommend acceptance.

**Key Questions For Authors:**

1. How should k be chosen for fDBD?: The paper shows fDBD is fairly stable, but the best version still depends on selecting top-k alternative tokens.

2. How general is the analytical proxy?: The proxy works well here, but it is still worth asking how it behaves across more model architectures.

3. Does it transfer to open-ended generation?: The paper shows strong results on reasoning benchmarks, but not yet on broader factual generation tasks.

**Limitations:**

yes

**Strengths And Weaknesses:**

**Strengths**
- Analytical proxy for feature mean: The paper derives a closed-form substitute for the training feature mean from language head parameters, avoiding direct access to LLM training data.

- LLM-specific adaptation of OOD detectors: It reformulates NCI and fDBD for token generation rather than applying classification-time OOD scores without modification.

- Simple sequence-level aggregation: It turns token-level uncertainty into a response-level score by averaging across decoding steps, making the detector easy to use.

**Weakness**
- Extra tuning for the best fDBD results: The strongest reported fDBD variant depends on choosing k on a training subset, so the top result is not fully hyperparameter-free.

- Limited analysis of why methods differ by setting: The paper reports that fDBD usually beats NCI and that selected-k helps, but it gives little deeper analysis of why this changes across datasets or models.

- Reasoning-focused evaluation scope: The experiments are strong for CSQA, GSM8K, and AQuA, but the paper does not test broader open-ended factual generation settings where hallucination detection is also important.

---

> ### Author Rebuttal · Authors · 2026-03-29
>
> Thank you for your support and valuable feedback. Please see our responses below.  We will include the additional experiments in the revision.
>
> **Weakness 1 and Question 1: hyperparameter selection**
>
> The best hyperparameter k is selected based on the validation set (line 369-377).
>
> **Weakness 2: method strengths differ by setting**
>
> Regarding why NCI can occasionally outperform fDBD, our intuition is that sometimes the unique manifold structure of a specific dataset—relative to the model’s language head—better aligns with NCI's inductive bias.
>
> Regarding why k selection may not improve over k=All: Our intuition is that k=All is favored when the features reside at similar distances to all alternative tokens. Such weak separation in distances makes top-k selection prone to discarding distances with only slightly worse uncertainty information. We explore this in Table 1 using a "distance ratio" (mean distance to the top-100 least likely vs. 100 most likely tokens) on Qwen-2.5-7B-Instruct.  A smaller ratio indicates that the distances to alternative tokens are more similar overall.  Consistent with our intuition, we observe that k=All is favored when this ratio is smaller.
>
> |  Dataset | Distance Ratio | Selected k |
> |---------|---------------|------------|
> | CSQA    | 1.64          | 100        |
> | AQuA    | 1.45          | 100        |
> | GSM8K   | 1.35          | All        |
>
> Table 1: k=All is favored when distances to alternative tokens are more similar (lower distance ratio).
>
> **Weakness 3 and Question 3: Generalizability to Open-ended Task**
>
> Following your suggestion, we experimented on the open-ended TruthfulQA dataset, reporting AUROC on Llama-3.2-3B-Instruct in Table 2. fDBD and NCI maintain competitive performance compared to single-sample baselines, validating generalizability to open-ended tasks.
>
> | Method | AUROC |
> |----------------------------|--------|
> | Predictive Probability | 51.32 |
> | LN Predictive Probability | 50.38 |
> | max(P) | 60.62 |
> | Perplexity | 63.78 |
> | COE_r | 48.91 |
> | COE_c | 55.16 |
> | fDBD | 67.83 |
> | NCI | 68.17 |
>
> Table 2: NCI and fDBD maintain competitive performance on TruthfulQA, validating generalizability to open-ended tasks.
>
>
> **Question 2: Generalibility of the Analytical Proxy**
>
> Following your suggestion, we experimented with alternative architecture Mistral-7B-Instruct-v0.3, reporting AUROC on CSQA in Table 3.  fDBD and NCI maintain competitive performance compared to single-sample baselines, validating generalizability of our analytical proxy to alternative architectures.
>
> | Method | AUROC |
> |----------------------------|--------|
> | Predictive Probability | 58.82 |
> | LN Predictive Probability | 58.75 |
> | max(P) | 57.75 |
> | Perplexity | 58.96 |
> | COE_r | 49.82 |
> | COE_c | 62.34 |
> | NCI | 64.91 |
> | fDBD | 66.36 |
>
> Table 3: NCI and fDBD maintain competitive performance on Mistral-7B-Instruct-v0.3, validating generalizability to alternative architectures.

---

> > ### Author Rebuttal · Reviewer_2t8v · 2026-04-04
> >
> > Thank you for sharing the strong results across multiple settings. All concerns have been resolved.

---

> > > ### Author Response · Authors · 2026-04-06
> > >
> > > We are pleased to learn that your concerns have been resolved. Thank you again for your constructive feedback.

---

### Official Review · Reviewer_w62R · 2026-04-01

**Soundness:** 2
**Presentation:** 3
**Significance:** 3
**Originality:** 3
**Overall Recommendation:** 3
**Confidence:** 3

**Summary:**

The paper studies hallucination detection in large language models (LLMs) through the lens of out-of-distribution (OOD) detection. The authors adapt two geometric OOD detectors (NCI and fDBD) to the autoregressive setting of LLMs, proposing (i) an analytical proxy for estimating training feature statistics without access to training data, (ii) a restriction of decision-boundary computations to a subset of likely tokens for efficiency, and (iii) a sequence-level scoring method based on averaging token-level uncertainty. The approach is training-free and operates on single samples. Experiments on commonsense and mathematical reasoning benchmarks allows authors claim consistent improvements over several baselines across multiple models.

**Compliance With Llm Reviewing Policy:**

Affirmed.

**Final Justification:**

The paper presents an interesting and timely idea with promising empirical results, but the current version relies on insufficiently justified assumptions and provides limited empirical evidence for its strongest claims. Strengthening the theoretical grounding, expanding baseline comparisons, and providing more rigorous experimental validation would significantly improve the work.

**Key Questions For Authors:**

1. **Validation of the analytical proxy**
	Can the authors empirically compare the proposed analytical proxy for the feature mean with the true empirical mean (computed on a representative subset of data)?
	A positive result would strengthen the theoretical justification of the method.
 2. **Overconfident hallucinations**
	How does the method perform in cases where the model produces incorrect answers with high confidence?
	This is a critical regime for uncertainty-based detectors and could significantly affect the conclusions.
 3. **Baseline completeness**
	Can the authors include comparisons with representation-based or internal-state methods (e.g., Mahalanobis-style detectors or the cited INSIDE)?
	This would clarify the relative advantage of the proposed geometric approach.
 4. **Multi-sample baseline configuration**
	The multi-sample baselines (SelfCheckGPT, Semantic Entropy) are evaluated with a fixed budget of three samples at temperature 0.5. How sensitive are their results to this choice? Would a larger sample budget narrow or close the observed performance gap?
 5. **Statistical significance**
	Can the authors report variance or confidence intervals for all experiments and verify whether the reported improvements are statistically significant?
	This would clarify whether the observed gains are robust.

**Limitations:**

No. The paper would benefit from a more explicit discussion of its limitations, particularly:
 - reliance on assumptions about feature geometry and mean estimation,
 - potential failure modes such as overconfident hallucinations,
 - sensitivity to decoding strategies and token selection,
 - absence of released code and datasets, which limits reproducibility.

**Strengths And Weaknesses:**

### Strengths

The paper addresses a timely and important problem: hallucination detection in LLMs, particularly in reasoning settings where existing methods often struggle.
The focus on training-free, single-sample detection is practically relevant, as it avoids both training overhead and expensive multi-sample inference.
Also, the connection between hallucination detection and OOD detection is conceptually appealing and could "inspire further research" (cit.) in uncertainty estimation for LLMs.

In particular, the work introduces an interesting reinterpretation of hallucination detection as OOD detection, and propose an adaptation of geometric OOD methods (NCI, fDBD) to the token-level prediction setting.
In this context, the idea of using an analytical proxy for training statistics and restricting computation to top-k tokens are reasonable design choices tailored to LLM constraints.

The proposed empirical evaluation definitely strenghten the contribution, although being somewhat limited (see below).
In particular, performance shows consistent, albeit moderate, improvements and the hyperparameter sensitivity analysis suggests some degree of robustness to the choice of k.

### Weaknesses

The most significant concern lies in the theoretical grounding of the method's core assumptions.
The claim that the language head parameters "implicitly reflect the geometry of the training feature distribution" is central to the approach, yet it remains insufficiently justified.
While classifier parameters are naturally influenced by feature representations through the training objective, it does not follow that they encode recoverable global geometric statistics such as the feature mean (particularly since the language head is trained discriminatively and multiple feature distributions can induce identical optimal classifiers).
Relatedly, the assumption that a zero-bias output layer implies a zero feature mean rests on implicit symmetry conditions over the hidden representations, something that is not correctly validated.
As a result, the proposed analytical proxy for the feature mean reads more as a pragmatic heuristic than as a principled estimator, and the paper would benefit considerably from either a formal justification or an empirical comparison against the true empirical mean computed on representative data.
Similarly, the claim in Section 3.4 that "we empirically validate these two uncertainty measures in the context of hallucination detection in Figure 1" overstates what Figure 1 actually shows: the figure provides a single illustrative example of the procedure rather than a systematic validation, leaving it unclear at that stage of the paper on what basis the reader should trust the appropriateness of the proposed metrics.

The empirical evidence, while encouraging, does not yet rise to the level needed to support the paper's strongest claims.
Early results in Tables 1 and 2 show relatively small performance differences on the order of 1--3 AUROC points, actually without any reported variance or statistical significance; this makes it difficult to assess whether the gains are meaningful or artefactual.
Some observed trends, such as the dependence on the number of retained tokens k, are explained qualitatively as a balance between informative and noisy tokens, but this pattern could equally be attributed to a generic bias--variance trade-off rather than a property specific to the proposed method.
The discussion of robustness under stochastic decoding is intuitively reasonable, yet the assumption that misaligned tokens are short-lived and do not significantly affect sequence-level scores is not theoretically justified and may not hold in more complex generation settings where errors propagate.
Table 3 does report mean and standard deviation over five random seeds, which is appreciated, but the relatively large observed variance and the limitation to a single dataset and model mean the evidence supports usability rather than the stronger claim of robustness.
A similar level of rigour (i.e., reporting variance across runs) is notably absent from the remaining experiments, including the main results in Table 4.

The baseline comparison also leaves room for improvement.
Several closely related approaches are absent: most notably representation-based or internal-state methods such as Mahalanobis-style feature-space detectors and INSIDE (Chen et al., 2024, ICLR), a work that the authors themselves cite but do not include in the experimental comparison, despite its direct relevance to the geometric perspective adopted here.
Additionally, multi-sample baselines such as SelfCheckGPT and Semantic Entropy are evaluated under a fixed and relatively small sample budget (three samples at temperature 0.5), which may underestimate their performance.
No sensitivity analysis to this choice is provided.
Including these comparisons (to the least) and reporting their sensitivity to configuration choices would clarify the relative advantage of the proposed approach.

Finally, the paper does not analyse the regime of overconfident hallucinations, where the model assigns high probability to incorrect outputs (see also INSIDE paper) -- an important failure mode for any uncertainty-based detector, and one that could materially affect the conclusions drawn.

Finally, the paper does not analyse the regime of overconfident hallucinations, where the model assigns high probability to incorrect outputs (see also the INSIDE paper):
this represent an important failure mode for any uncertainty-based detector, and one that could materially affect the conclusions drawn.
On a related practical note, the authors do not release code or processed datasets.
For a method whose gains are moderate and whose core assumptions would benefit from independent scrutiny, the absence of a reproducibility package makes it difficult to fully assess the contribution and limits the paper's potential impact on follow-up work.

#### Minor Details
The formulation of the hallucination detector in Equation 1 adopts a binary framing (hallucination vs. non-hallucination) with a scoring-plus-threshold mechanism.
While this abstraction is reasonable, it is worth noting that recent work has shown that individual responses can contain both hallucinated and non-hallucinated segments, and that hallucination severity is better captured as a spectrum rather than a binary label.
The authors would benefit from acknowledging that the proposed formulation represents one possible operationalisation rather than the only one.
Separately, in Section 3.1 the language head is described as a linear classifier, which is true for the vast majority of current architectures but is stated as a universal property; a brief qualification (e.g., "typically linear") would improve precision.

---

### Decision · Program_Chairs · 2026-04-30

**Decision:**

Accept (regular)

**Comment:**

This paper studies hallucination detection in LLMs from a geometric OOD perspective, adapting NCI and fDBD into training-free, single-sample detectors for token generation. The reviewers’ scores are 3, 4, 4, and 5. Overall, I recommend weak accept. The paper offers a clear and timely perspective by connecting hallucination detection with OOD detection, and the empirical results are consistently strong on reasoning benchmarks. Importantly, the rebuttal substantially strengthened the paper: the authors added evidence on open-ended generation, alternative architectures, MoE models, model scaling within the same family, instruction tuning, and inference latency. Three reviewers indicated that their concerns were fully resolved, while the remaining concerns are relatively narrow and do not outweigh the paper’s practical value and conceptual novelty. In my view, this is a solid contribution that is likely to stimulate follow-up work in LLM uncertainty and hallucination detection.